# Abrahamic Family or Start-Up Nation?: Competing Messages of Common Identity and Their Effects on Intergroup Prejudice

**DOI:** 10.3390/bs15040460

**Published:** 2025-04-03

**Authors:** Tsafrir Goldberg, Laila Abo Elhija Sliman

**Affiliations:** Department of Learning and Instructional Sciences, University of Haifa, Haifa 3498838, Israel

**Keywords:** interfaith attitudes, intergroup relations, common ingroup identity theory, media representation of religion, Start-Up Nation, interfaith education

## Abstract

Just as Israel brands itself as a progressive “Start-Up Nation”, Israeli citizens increasingly identify as religious. Religion plays an increasing role in intergroup tensions in Israel. Negative effects of religion and its public representations on intergroup attitudes are well researched, but little is known of its positive effects. We ask whether religion can also play a role in improving intergroup attitudes. This study explores the effects of three different public and media representations of shared identity and tolerance on interreligious prejudice among Israeli Muslim adolescents and young adults. The interventions included an interfaith similarities-based common ingroup identity (focusing on shared aspects of Judaism and Islam), a modern national universalistic approach (focusing on religious tolerance), and a modern academic technological identity (highlighting Israel as a “Start-Up Nation”). Findings indicate that the interfaith similarities-based intervention had the most substantial impact in reducing prejudice, specifically by decreasing stereotypes and increasing willingness for social encounters. In contrast, the national universalistic and technological identity interventions were less effective, and in some cases even increased perceived threat or failed to mitigate stereotypes. These findings highlight the potential for leveraging interfaith commonalities as a foundation for intergroup prejudice reduction.

## 1. Introduction: Religion and Intergroup Relations

In the last decade, religion appears to play an increasing role in tensions and conflicts between Israel and the Palestinians, as well as within Israel, between its Jewish and Muslim citizens. Can religion also play a mitigating role? This article presents two studies exploring the effect of religion and an intervention focusing on interfaith commonalities of traditions on prejudice reduction, compared to interventions stressing more modern outlooks, focusing on universalistic equality of rights or on academic technological collaboration.

### 1.1. Negative Impact of Religion on Intergroup Relations

With the growing power of the Hamas (Arabic acronym for Islamic opposition movement) in Gaza and the west bank and with the growing influence of fundamentalist and ultraorthodox parties in the Israeli polity, it appears that religion plays an increasingly polarizing role intensifying conflict and extremism ([44]; [53]). Within Israel, mounting tensions stem from Islamic movements’ claims that the Al-Aqsa Mosque is in danger, and from the adversarial demands of Jewish fundamentalists for renewal of Jewish rituals on the temple mountain ([33]). The unprecedently widespread and violent protests by Arab Israeli citizens in May 2021, leading to Jewish-Arab riots were, at least declaratively, centered on this religious issue (although they may be interpreted as stemming from social and economic tensions) ([21]). While many Israeli Muslims support intergroup relations, those identifying as more religious report less connection with Jews, a disconnect which usually predicts negative intergroup attitudes ([2]). Since an increasing share of Israeli youths, both Muslim and Jewish, self-identify as religious, even among sectors assumed to be secular ([34]), the influence of religion on intergroup relations would probably increase for better and for worse.

Studies in Israel and abroad have shown religiosity to be a predictor of prejudice towards outgroups and members of other religions ([3]; [7]; [45]). Exposure to religious symbols of an outgroup’s religion was reported to arouse prejudice, as evidenced by the rise in negative stereotypes among (Israeli Muslim, though not among Jewish) college students ([43]). This may be because believers tended to assume the holy writings of the outgroup stress negative attitudes towards their own group ([42]). Negative representations of outgroup religions abound and are frequently based on religions’ traditional prejudices. [52] ([52]) integrated theory of threat (ITT) assumes negative cognitive representations, such as religious stereotypes would lead to feelings of intergroup (in this case, interfaith) threat. Intergroup threat leads in turn to aversive behavior and preference for interfaith distancing. To sum, we may expect religion and exposure to its representations to increase prejudice and impede intergroup relations. However, can religion also play a positive role in intergroup relations and encourage positive intergroup attitudes?

### 1.2. Potential Positive Impact of Religion

As [20] ([20]) and [1] ([1]) claim, religions can serve as a basis for intergroup understanding and compassion and interreligious conflict is historically a derivative of political tensions rather than vice versa. [53] ([53]) in his analysis of the role of religions in the Israeli–Palestinian conflict contends that peace negotiators missed the potential of Jewish and Muslim religious sentiments and moderate strands as a deep-seated basis for enduring peace. Acknowledging religious values of believers should allow for self-affirmation and mitigate defensiveness in intergroup transactions ([10]). [48] ([48]) have shown that both Muslim Palestinians and Israeli Jews surprisingly tend to assume that outgroup members’ thinking about God encourages intergroup benevolence. Even feelings of intergroup threat did not impede this effect of religious reference on prosocial attributions. These positive effects of thinking about the other’s religion may increase with acquaintance with the other religion, as offered for example by interfaith education.

Interfaith education, the study of other religions and their believers, sometimes with members of other religions, is considered to encourage interreligious understanding and improve intergroup attitudes and relations ([5]; [24]; [29]; [47]). Learning with believers of the other religion had prejudice-reducing effects as predicted by the contact hypothesis. Even just learning more about other religions had positive effects on intergroup attitudes, especially when it included counter stereotypic examples ([32], [31]). Studying about the tolerance, rich culture and advances in science under the medieval Muslim regime, as well as Muslim–Jewish interreligious influences (as part of a mandatory Israeli inter-cultural curriculum), reduced Jewish students’ Islamophobic stereotypes and increased their willingness for encounter ([18]). Moreover, while religions are usually considered to enhance intergroup boundaries, they may open trajectories for more inclusive overarching common ingroup identities. This can reframe intergroup social categorizations, which usually demarcate and distance groups ([7]).

### 1.3. Religion and the Common Ingroup Identity Theory

The Common Ingroup Identity theory assumes that such re-categorization into a shared group reduces competitive social comparison and its derivative outgroup derogative representations, such as negative outgroup stereotypes ([16]). Thus, for example, acknowledging Judaism and Islam as part of the “Abrahamic tradition”, predicted positive outgroup attitudes and willingness for contact towards members of the other religion among Jewish and Muslim respondents ([25]). Perceived common identity can be also promoted by stressing moral similarities among the members or the history of two separate, even adversarial groups ([8]; [50]). However, common identity may also challenge ingroup members need for optimal distinctiveness, threatening the sense of uniqueness giving positive meaning to group membership, especially for minority members ([27]). This may explain the negative relation of religiosity and intergroup contact on Israeli Muslims ([2]). Hence, it is sometimes recommended to rely on a dual-identity approach, in which the superordinate common identity acknowledges the existence of distinct sub-identities (as in the idea that Islam and Judaism are two distinct members in the “Abrahamic family of religions”) ([38]). This may be especially relevant to groups which acknowledge shared mores and interests but principally resist assimilation, as in the case of Jewish and Muslim religious minorities, which harbor an ambivalent or restrained stances towards a national common identity ([30]).

In previous studies we found that studying interreligious similarities of Judaism and Islam indeed reduced Jewish students’ stereotypes and increased willingness for contact ([19]). However, the effect on Muslim students was insignificant or inverse. We interpreted the finding as a reaction to the threat interfaith similarities posed to Muslim minority optimal distinctiveness ([18]). Still, this threat should also have affected the Jewish students, who are in fact a small Jewish enclave in an overwhelmingly Muslim region, in a country setting a high value on group distinction. We therefore sought to replicate and further investigate the effect interreligious similarities had on Muslim students.

#### Study 1

The results of the previous study, showing that unlike Jewish students, Muslim students studying about the similarities between Islam and Judaism increased prejudice, seemed quite surprising. Therefore, we focused this current study on Muslim students, trying to replicate the effect. However, before replicating the study, we sought to account for the negative effects in additional ways. Learning materials teachers produced in the previous study ([18]) consisted of a text noting the parallels between central religious duties in Islam and Judaism, and a text comparing Muslim and Jewish policies of tolerance towards other religions. Muslim students were requested to write short reflections following the study, and we turned to those for some additional insights. As we analyzed students’ reflections, we noted that most of them claimed the Muslim medieval religious tolerance code (“Umar’s Pact”) was fairer than the attitude to minorities in Israel’s declaration of independence.

Both texts are commonly shown in public places and communicate a declarative stance towards minorities. “Umar’s Pact”, which sets the protected though inferior status of (Christian and by inference, Jewish) monotheistic minorities under Muslim rule, is one of the frequent results on web searches referring to positive Muslim relations with Christians and Jews[note 1]. Israel’s declaration of independence is shown in Israeli institutions and classrooms (though infrequently in Arab schools), offering a seemingly more universally egalitarian approach to minorities than the “Umar Pact”[note 2]. Muslim students’ perception of it as less fair was possibly aroused due to the prominence of a Jewish-Israeli identity in the declaration. Additionally, it may be due to the allusion to the conflictual context of the 1948 war.

We therefore sought to distinguish between the effect of a solely religious intervention, assumed to arouse a sense of common identity based on the similarities of religious edicts, and an intervention making an analogy between past and present, assumed to arouse a shared identity based on the similarity between the Muslim empires’ declared religious tolerance and present-day Israel’s declared freedom of worship and equality for believers of all religions.

Research question:

What are the effects of an interreligious similarities intervention versus that of an analogy between a medieval Muslim and contemporary Israeli policy towards minorities, on Israeli Muslim adolescents’ interreligious prejudice and willingness for social closeness?

Hypothesis

Based on previous research, we hypothesized that an interreligious similarities intervention would have a stronger prejudice-reduction effect than that of an analogy between a medieval Muslim and contemporary Israeli policy towards minorities. This is because, while overtly universalistic and inclusive, national implications of the minority policies analogy as well as its conflictual context are assumed to arouse associations of Israeli Muslims as a threatened minority and prime religious identity in a negative attitudinal direction ([3]; [33]; [44]).

## 2. Materials and Methods

### 2.1. Participants

The previous study focused on seventh grade students (the grade in which Israeli History curriculum instructs teachers to discuss the Relations of Judaism and Islam). Hence, participants were 324 seventh grade students (220 female) from five Arab speaking schools serving towns and villages in northern Israel with an almost totally Muslim population (typical of patterns of settlement and regional planning in Israel’s Arab small townships). A total of 38% of participants identified (or viewed their family) as religious, 46% as conservative, 5% as secular, and 10% declining to respond (a slightly higher proportion of religious than shown in [34] ([34]) youth survey). Informed consent was obtained from parents of all subjects involved in the study.

### 2.2. Procedure

Participants were placed into three conditions: The first, a solely religion-focused intervention, stressed commonality of major religious edicts in Islam (the “Pillars of Islam”) and Judaism. The second, an analogy to present day acceptance of minorities, traced the similarity between tolerance towards minorities in the Muslim empire (’Umar’s pact’) and in Israel (Israel’s independence declaration); and the third served as control (a regular general history lesson). In all conditions, students listened to a teacher present the topic and completed a worksheet (see Appendix A) for the duration of a full lesson (45 min).

Prior to studying the topic of the lesson, students completed a questionnaire tapping interreligious attitudes based on a three component model of attitude ([39])—cognition (stereotypes); affect (threat); behavior (willingness for social closeness/distance) adapted by the author, translated into Hebrew and Arabic ([18]), and validated by the second author, a native speaker, based on interreligious attitudes questionnaires ([14]; [23]; [26]) (see Appendix A). They filled in the same questionnaire again within a week of intervention lesson. The study was approved by University of Haifa ethics committee and Israeli ministry of education.

### 2.3. Variables

**Stereotypes**: This measure consisted of nine items presenting a spectrum between a monolithic negative view and a more complex diverse description of Jews (e.g., “Compared with other people, Jews are uncivilized and backward (5)—Jews are as civilized and advanced as other people (0)”, Cronbach’s α = .79).

**Threat**: This measure consisted of eight items presenting perceived threat of Judaism and Jews (e.g., “If Jews serve as care takers of Muslims, they will harm the Muslims” and “Jewish hospital workers save the lives of many people”—reverse coded. Cronbach’s α = .69). Agreement with items was recorded on a six-point Lickert-type scale with higher number representing higher threat. Reverse items were inverted.

**Social Distance/Closeness**: This measure consisted of ten items representing a negative or positive affective behavioral stance to Jews (e.g., “I would not live in the vicinity of Jews” or “I would be interested to speak with Jews”). Agreement with items was recorded on a six-point Lickert-type scale. Since exploratory factor analysis showed negative and positive items did not load on the same factor and scale reliability was low, we chose to maintain them as two separate variables, leading to moderate scale reliability (Social Distance Cronbach’s α = .69; Social Closeness Cronbach’s α = .77).

Participants also filled in demographic questions as to gender and degree of religiosity (on a three-point scale ranging from secular through conservative to observant Muslim).

## 3. Results

As we can see from Table 1, all study variables are weakly to moderately correlated. We should also note that religiosity is positively correlated with threat and negatively correlated with social closeness, as can be expected according to current knowledge.

To compare the effects of the solely religious similarities intervention with the effects of the intervention using an analogy to present day acceptance of minorities on the various aspects of prejudice, we performed a multivariate analysis of variance over change scores for all variables (computed by deducting pre from post intervention score) with condition as the between-subjects factor. The analysis showed a significant multivariate effect (F(8,638) = 6.68, *p* < .001, ƞ2 = .08). Significant univariate effects were found for all variables except change in social distance (see Table 2). As can be seen, in the solely religious similarities condition, stereotypes and threat decreased and willingness for social closeness increased, indicating altogether a reduction of prejudice compared to the two other conditions (except for the reduction of stereotypes in the control condition). By contrast, in the medieval to present-day analogy condition, changes were in the opposite direction, if they occurred at all. This finding supports the hypothesis that an interreligious similarities intervention was more suited to reduce prejudice among Muslim Israeli adolescents, than an intervention relating to the universality of Israeli polity, hence to a common national identity.

To trace possible interaction between experimental condition and degree of religiosity an additional MANOVA was performed over all change variables with condition and religiosity as between subject factors (we report this test separately as we omitted non-responders and the few seculars to create statistically robust comparisons. This significantly reduced the sample). Test revealed significant multivariate effects for condition and religiosity (F(8,532) = 5.96, *p* < .001, ƞ2 = .08 and F(4,265) = 3.70, *p* < .01, η2 = .05, respectively). No significant interaction effect was found; hence, condition did not moderate the effect of religion or vice versa. Conditions had similar univariate effects as above. Univariate tests revealed a significant difference between religious and conservative participants. Religious participants showed a decrease in threat (M(SD) = −.10(.88)) and increase in social closeness (M(SD) = .39(1.20)) while conservative participants showed opposite or negligible change (M(SD) = .11(.97) and (M(SD) = .08(1.10), respectively). This phenomenon of religiosity apparently encouraging prejudice reduction seems to stand in contrast with the initial findings in which religiosity was positively associated with threat while negatively associated with willingness for social closeness.

We fitted a hierarchical regression model predicting change in social closeness from religiosity, condition (dummy variable representing interreligious similarities vs. two other conditions), change in stereotypes and in threat. The model was significant (F(4,287) = 11.80, *p* < .001, adjusted r^2^ = .13) predicting 13% of the variability in social closeness, with each predictor adding significantly to the predictive strength of the model, and all coefficients showing a significant association with social closeness, as can be seen from Table 3. Partial correlations remained similar to initial coefficients, indicating no mediation effect of religiosity.

## 4. Discussion

Findings of study 1 show that reference to interreligious similarities has a stronger potential to reduce interreligious prejudice and anxiety than a reference to the contemporary Israeli declaration of religious equality and religious freedom (which led to increased stereotypes and threat). This runs somewhat in contrast to the assumption that the counter-stereotypic information in both interventions would reduce stereotypes about Jews and Judaism and enhance social closeness ([32]). It also counters the assumption both interventions would foster common identity and consequently a reduction of prejudice as they stress moral similarities such as religious tolerance ([8]; [16]). Our interpretation of these effects is that a message focusing on interreligious similarities may have indeed helped arousing a sense of common interreligious identity, a shared membership of “Abrahamic religions” ([25]). The equal footing of both religions in the comparison and their clear boundaries satisfied optimal distinctiveness needs ([27]). This common identity may have led to a decrease in intergroup boundaries; hence, a lower need for social comparison and denigration of the other ([16]). This led to reduced negative stereotyping of outgroup members, which according to [52] ([52]) integrated theory of threat may have reduced perceived threat and increased willingness for social closeness.

On the other hand, the comparison between Umar’s pact, representing the Muslim empire’s religious tolerance towards monotheistic minorities, and the Israeli declaration of independence clauses on religious equality and freedom may have in fact aroused a sense of demarcation and possibly threat. We assumed that as [48] ([48]) have shown, reference to religious authorities (such as the Khalif or the Jewish prophets’) justice and compassion would arouse the perception that religion encourages believers to show intergroup benevolence. However, while delivering an overt universalistic message, supposed to assure a common identity, the comparison denotes the status of Muslims as a minority, alongside implying that Israeli national identity is tied to Jewish identity and raising the memory of violent conflict ([44]). Thus, social categorization may have activated social comparison and led to increase in stereotypes, and consequently to increase in threat perceptions and a decrease in willingness for social closeness. It may also be that mention of Jerusalem in the Umar pact as well as the name of the Khalif Umar, founder of the Al-Aqsaa Mosque may have aroused negative associations to the constant Arab media messages about Jewish threats to the Muslim holy site ([33]). It is also possible the universalistic national approach may represent in itself an anathema to Israeli Muslims who view boundaries also as a safe guard ([2]; [30]), and hence challenged the need for optimal distinctiveness ([27]).

The association of religiousness, reduction of threat and increase in willingness for social closeness appear to support claims the religion can promote intergroup relations ([7]; [20]). It seems probable that the more religious participants reacted more favorably to references to religious duties they knew and observed, arousing self-affirmation, and easing attitude change ([10]). However, it is unclear why they did not react more negatively to the mention of Umar’s pact with its current negative associations.

Hence, our findings point to the problematics of messages seemingly focusing on the universalistic aspects of Israeli national identity as a common identity to reduce interreligious prejudice, anxiety and contact avoidance. By contrast, the effects of the inter-religious similarities intervention, along with the positive association of religious observance with reduction of prejudice, seem to show a promising direction for work on religion as a basis for shared identity and prejudice reduction. This may be due to the wider space it affords for dual identity (Muslim and “Abrahamic”) within a common ingroup identity ([38]). However, since we did not measure common identity directly, it remains to be shown whether indeed the intervention led to an increase in common identity as a distinct construct and whether the latter affected the observed changes. We should also note that interreligious messages may not play to the taste of every Israeli-Muslim, especially among the more secular and among higher education graduates ([2]) and are not common in popular media. We therefore should strive to explore whether references to a common identity which is not overtly national but also not religious can have a similarly positive effect on shared identity and mitigate prejudice. To tackle these issues, we initiated study 2.

### Study 2

Following study 1 which showed the limitation of declaratively universalistic national messages as a common identity intervention, we sought to compare the effects of a religious common identity-based intervention to the effects of a non-religious but also non-national common-identity intervention. To achieve this, we turned to Israel’s growing self-identification as the “Start-Up NationStart-Up Nation”, at the core of which lies scientific and technological innovation achieved through collaborative teamwork. Although Israeli “techno-capitalism” is sometimes criticized as exclusionary, hi-tech teams are often cross denominational, transnational and even global or cosmopolitan in character. Due to the constant demand for employees in the Israeli hi-tech industry, it has, somewhat belatedly, opened new opportunities even for Arab-Muslim citizens ([17]; [28]). Israeli Government and NGOs actively encourage their entrance into science and technology tracks in higher education, and Arab high schoolers embrace “tech identity” and aspirations ([4]; [6]). Consequently, members of Arab and Muslim society in Israel have turned in growing numbers to Israeli Academe’s and the industry’s technological and scientific tracks ([22]). Thus, reference to intergroup collaboration within hi-tech and academic initiatives may appeal to young Muslim adults, arousing a sense of a shared “Start-Up NationStart-Up Nation” identity.

Representations of minority members’ professional roles in the media can impact intergroup attitudes, both positively and negatively. When minority members are shown in high status professional roles, or collaborating positively with majority members, viewers tend to improve their intergroup attitudes and may develop a sense of common identity ([59], [58]). As diversity in the workplace is increasingly seen as a positive attribute, images of diverse work teams may abound ([15]; [35]). Representation of Jewish–Arab collaboration in work teams has been used in Israeli media to transmit unifying messages and help overcome intergroup prejudice and threat. Media examples of collaboration were taken mainly from emergency health services, where integrated teams abound ([41]; [51]; [57]). While intergroup and interreligious relations at the workplace are not free from external and internal tensions and inequality even in Israel’s most diverse and integrated vocational field of health care, workers and outsiders tend to view it as a realm of collaboration and solidarity ([11]). Lately, the growing participation of Israeli Arabs in the hi-tech industry and academic tracks has also been represented in the media in various ways ([12]; [56]).

Hence, we chose representations of participation in scientific and hi-tech initiatives as a focus for a “Start-Up NationStart-Up Nation” common-identity intervention, to be compared with the interreligious similarities intervention, assumed to arouse an “Abrahamic identity”. This approach may be more relevant to young adults and adults rather than middle-schoolers. Hence, we chose to compare the common identity effects of both approaches among young adults and adults.

The two alternative common-identities appear to differ significantly, a religious identity is seemingly present and past oriented ([55]) while a technological scientific identity is future oriented ([37]). A scientific technological identity may seem at odds with religious identity ([13]). Hence, we may expect that students and graduates of academic tracks, especially young adults may be more attuned to a scientific technological identity approach. However, this perception may apply mainly to atheists, while conservative and religious individuals may see science and religion as compatible ([49]). As most Arab Israeli citizens, especially of Muslim decent, identify as conservative or observant believers, the scientific-religious identity dichotomy may not affect them. On the other hand, they may react more favorably to messages containing religious references ([2]; [9]; [46]).

The goals of this second study were to compare the effects of a religious common ingroup identity-based intervention to the effects of a non-religious but also non-national shared ingroup identity intervention on the sense of common identity, interreligious prejudice and willingness for social closeness. Furthermore, while in the previous studies we measured only interreligious attitudes and assumed that positive changes in them are the effect of increased common identity, here we propose to also tap changes in common identity directly and establish its relation to interreligious attitude change.

Research questions:

What are the effects of a religious common identity intervention versus a technological-academic common identity intervention on the sense of common identity, inter-religious prejudice and willingness for social closeness?

Does increase in interreligious common identity affect interreligious attitudes?

## 5. Materials and Methods

### 5.1. Participants

We chose a sample of adults since we assumed that the notion an academic profession based common-identity would strike a chord more strongly among participants of working age. We used a convenience sample of 137 Israeli Muslim citizens (80% women) aged between 18 and 50 (60% 18–30) who responded to postings in social media. A total of 93% of participants had or were studying for an undergraduate degree. A total of 69% of them studied in an integrated Hebrew speaking institute alongside Jewish students. And 75% identified as conservative (moderately observant) Muslims, 14% as religious (observant) Muslims, and 11% as secular.

### 5.2. Procedure

Participants were recruited online via social media and university networks. Once they joined the study, they filled in an interfaith attitudes’ questionnaire like that of study 1, with additional 4 questions comprising a shared-identity scale based on [25] ([25]).

Two weeks after completing the questionnaire participants were directed to an online form in which they read excerpts according to experimental condition and answered comprehension and opinion questions to ensure on-task engagement. Upon the completion of the task (average duration 25 min) participants were directed through a new link to a questionnaire similar to the one they filled out before, at the end of which they received a link to a refreshments’ coupon worth $10.

Experimental conditions consisted of three groups. the first was an interreligious similarities intervention like that described in study 1, in which participants matched the “five pillars of Islam” edicts to parallel Jewish edicts based on popular media representations[note 3] and responded to the question whether the two religions should be seen as “sisters” and whether similarities are supposed to increase interfaith trust in Israel. The second condition was a “Start-Up Nation” identity intervention, in which participants read about the rise of Israeli hi-tech industry and the state’s support of Arab Israeli citizens’ (especially of Arab women’s) entry into technological academic professions, taken from several media reports[note 4]. The third condition which served as control, focused on herbal medicine, was adapted from a popular Arabic youth science site[note 5].

### 5.3. Variables

Common identity: This measure consisted of four items based on [25] ([25]) referring to a common “Abrahamic” identity on a five-point Leickert scale (e.g., ”You may say that Muslims and Jews belong to the same ‘family’ of religions” Cronbach’s α = .90).

The rest of the variables were like those described above, namely Stereotypes, Threat (without reverse items for reliability considerations), Social Distance and Social Closeness but on a five-point scale (respective Cronbach’s α’s = .84, .68, .49, .46)

## 6. Results

Table 4 shows means, SDs and bivariate correlations for all research variables. As we can see, religiosity was negatively associated with a sense of shared identity and positively related to perceived threat of Jews. As study 1’s initial results showed, and previous studies lead us to expect, religion may be seen as impeding interfaith relations.

To answer the first research question, we first sought to see whether the condition had a differential effect on changes in research variables; then, we checked whether a significant change occurred within each condition. We computed change scores for each variable, subtracting the pre intervention from the post intervention score (see Table 5). Unlike study 1 findings, religiosity had no significant correlation with changes in any of the research variables.

A MANOVA performed over change scores with the condition as between subjects variable revealed a significant multivariate effect (F(10,262) = 1.93, *p* < .05, η2 = .07). A univariate effect appeared for the condition over threat (F(2) = 7.61, *p* = .001, η2 = .10) with pairwise comparisons (using Bonferroni correction) showing a significant difference in change directions between the control and both experimental conditions, as threat increased in the former while it decreased in the latter.

We then performed separate repeated measures, ANOVAs, within each condition to ascertain whether any experimental condition caused a significant change in research variables. As we can see from Table 6, both experimental conditions revealed a significant increase in common identity and decrease in various aspects of prejudice. In the interreligious similarities condition, stereotypes decreased significantly and willingness for social closeness increased significantly. In the “Start-Up NationStart-Up Nation” identity condition and the control condition, threat increased significantly. We should caution that though these changes are significant, they do not indicate a significant difference in effect between conditions.

In both experimental conditions, the intervention contained a direct question about the plausibility of a common identity (e.g., are Judaism and Islam sister religions in your opinion?/can Jews and Arabs in Israel be seen as part of a common “Start-Up NationStart-Up Nation”?). Responses differed between the interreligious similarities condition where 87% of participants responded in a clear affirmative, 13% negative and the “Start-Up NationStart-Up Nation” condition, in which 64% affirmed, 18% responded “maybe” and 18% negative (Chi^2^(2) = 12.51, *p* < .01). This contrasts with the finding that the effect on common identity in the “Start-Up Nation” condition was slightly higher (explaining 11% vs. 9% of variability) and had higher effects.

To answer the second research question as to whether a change in interreligious common identity affects interreligious attitudes, we fitted a structural equations model (using AMOS27 software, see Figure 1). Guided by integrated threat theory ([52]), the model assumed a change in shared identity impacted a change in stereotypes and social closeness and indirectly affected a change in threat through a change in stereotypes. The model showed a decent fit (Chi^2^(2) = 1.29, *p* = .26; NFI = .97, CFI = 99; RMSEA = .05)[note 6], accounting for 19% of the variability of change in social closeness, 7% of the variability of change in stereotypes, and 5% of the variability of change in threat. We performed bootstrapping which revealed the change in common identity had a significant indirect effect on the change in threat and social closeness (beta’s = .06, .07, *p*’s < .05). Change in threat had no significant impact on social closeness (contrary to ITT assumptions, although it was associated with social distance). This may help explain why in the interreligious similarity condition, wherein stereotypes decreased significantly, we also find an increase in the willingness for social closeness, and why such a process did not occur in the Start-Up Nation condition where stereotypes hardly changed. However, it does not fully explain why in the “Start-Up Nation” condition, although common identity significantly increased, stereotypes did not decrease, nor did social distance although threat decreased.

## 7. Discussion

Study 2 shows the potential of both the interreligious similarities messages and the participation in hi-tech world messages to promote a sense of common identity between Jews and Muslims. These results support prior research about effects of interfaith education and encounter with interreligious similarities ([19]; [40]). They also point to the beneficial effect of media representations of collaboration within diversity in academy and industry, especially in the globalizing and transnational realm of the advanced technological corporate world ([15]; [54]; [59]). Increase in common identity in both experimental conditions was significant (although it is hard to isolate the effect of intervention, as common identity change did not differ significantly from the control condition). It appears to have spurred change in various aspects of interreligious attitudes as assumed by common identity theory, as social categorization is dimmed and the need for outgroup derogative social comparison decreases ([16]; [36]). It should be noted that although most participants identified as conservative or religious, religiosity did not impact the positive reactions to science-focused, future-oriented, “Start-Up Nation” messages.

It is worth noting, however, that the two types of intervention had different effects on the cognitive, affective and behavioral aspects of intergroup attitudes. Engagement with interreligious similarities messages appears to drive change both in the cognitive aspect of intergroup relations (stereotypes) and in the behavioral aspect (willingness for social closeness). This may be because the interreligious similarities messages activate a more ingrained and accessible set of associations between Judaism and Islam related to traditions all Muslim respondents are acquainted with (prayer, charity, etc.) ([2]). Such traditions may more easily be related to notions of moral similarities, which as [8] ([8]) claim, enhance common identity and counter negative stereotypes (by stressing aspects like charity and moderation). The perception of moral similarity within two distinct traditions can also legitimize closer social interaction, perhaps mitigating fear of total boundary dissolution. The effect of religious intervention held regardless of a lack of significant change in perceived threat as noted also by [48] ([48]).

By contrast, “Start-Up Nation” messages may invoke more competitive associations, stemming from the nature of techno-capitalism as well as from the less inclusive characteristics of the Israeli hi-tech scene ([17]; [28]), which did less to counter stereotypes. As hinted by respondents’ skeptical responses about viewing Jews and Arabs as part of a common “Start-Up Nation”. However, this skepticism should have also limited the effect on threat which decreased significantly in this experimental condition.

As the structural equation modeling showed, it is plausible to assume that arousal of common identity reduces stereotypes and increases willingness for an interreligious social encounter, and that its effect is also mediated by stereotypes onto threat. This aligns with [52] ([52]) integrated theory of threat, in which stereotypes drive threat and contact motivation.

This may also explain why engagement with “Start-Up Nation” participation messages, while apparently mitigating the negative affective aspects of interreligious attitudes (interreligious threat), did not lead to the expected change in the behavioral aspect (either as a reduction of preference for social distance or increase in willingness for social closeness). Viewing outgroup members as participating in the same realm of advanced research or industry may indeed reduce threats of violence or world domination. However, since stereotypes are not mitigated, the willingness for social interaction beyond the professional realm does not increase. Findings do not fully support [52] ([52]) ITT, for although stereotypes did impact threat and social closeness motivations, threat did not impact social closeness, nor did a significant reduction in threat lead to a reduction of social distance.

## 8. General Discussion

Both studies point to the potential of interventions communicating interreligious similarities to reduce intergroup prejudice. Tradition-oriented interventions appear to have stronger potential to reduce stereotypes and increase willingness for encounter than interventions with more modern orientation, stressing the advantage of current universalist equality of rights, or the opportunities opened by collaboration in technology and academe. These findings stand in contrast to the vast evidence on the negative impact of religion intergroup relations ([45]). They add to the small but growing body of evidence hinting that religion may play a positive role in intergroup relations, especially through exposure to interfaith religious and moral similarities ([7]; [18]; [32], [31]).

Findings of study 1 hint that, at least in the Israeli context, reference to the modernist and seemingly universalistic declarative equality of rights under the nation state, may increase intergroup prejudice among Muslim minority members. We interpreted it as indicating that the stress on Jewish aspects of the polity failed to arouse a sense of common identity, leading instead to defensive reactions arousing stereotypes and threat. A reaction also possibly enhanced by the threat the modernist universalist outlook posed to the more traditional Muslim identity ([30]). However, in study 2, we managed to show that both a presentation of similarities in tradition and media coverage of collaboration in technological innovation managed to significantly increase a sense of common identity between Muslims and Jews among Israeli Muslims. This may be because they offer space for dual identity, both the particularistic faith-based and the over-arching common identity, and do not challenge the need for optimal distinctiveness ([27]; [38]). Both approaches significantly mitigated aspects of intergroup prejudice such as stereotypes and threats. As the model tested above shows, increase in common interfaith identity appears to foster these changes, in line with the common ingroup identity theory ([16]). It remains for further research to explore why such changes led to a significant change in motivation for intergroup encounter only in the interreligious similarities condition.

Findings point to the potential of presenting such similarities and examples of collaboration in the media and public sphere to mitigate widespread interfaith tensions ([19]; [59], [58]). Emphasis on interreligious similarities should be considered a promising trajectory as Israeli youth increasingly identify as religious and religion achieves greater influence on social and political life ([2]; [34]; [44]).

The studies presented have several limitations. They are based on self-reports, which are germane to social pleasing effects (the fact that control groups also showed some prejudice reduction effects hints that simple exposure to the same questions may arouse a wish to appear less prejudiced). Additionally, the effects reported were small, and in some aspects did not significantly differ from the control, making it hard to isolate the effect of the intervention. In addition, the “Start-Up Nation” intervention was not applied to a comparable Jewish sample. Further research should seek to replicate these effects and to compare them to the effects on Israeli-Jewish adults.

## Figures and Tables

**Figure 1 behavsci-15-00460-f001:**
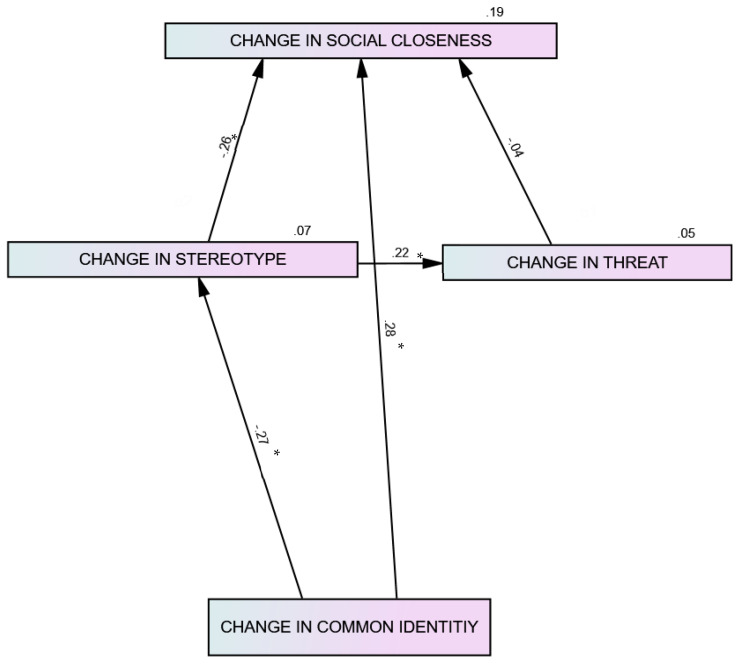
Structural equation model path diagram predicting change in willingness for social closeness from changes in common identity, stereotypes and threat. * *p* < .01. Percent of explained variability (r^2^) above variables.

**Table 1 behavsci-15-00460-t001:** Means, standard deviations and bivariate correlations of all research variables.

	M	SD	Religiosity	Stereotypes	Threat	Social Distance
Religiosity	2.45	.63				
Stereotypes	1.97	.74	−.12 *			
Threat	2.11	.89	.14 **	.48 **		
Social distance	1.79	1.09	.09	.31 **	.28 **	
Social closeness	2.53	1.29	−.22 **	.30 **	−.38 **	−.25 **

* Correlation is significant at the .05 level (2-tailed). ** Correlation is significant at the .01 level (2-tailed).

**Table 2 behavsci-15-00460-t002:** Change scores for research variables by condition.

Condition	Control (N = 154)		Religious Analogy (N = 60)		Present Analogy (N = 110)		F	η2
	M	SD	M	SD	M	SD		
Change in stereotypes	−.21 ^b^	.72	−.12 ^c^	.58	.36 ^bc^	.93	18.62 **	.10
Change in threat	.11 ^d^	.91	−.28 ^de^	1.08	.14 ^e^	1.00	4.06 *	.03
Change in social distance	−.21	1.37	−.28	1.08	−.01	1.31	1.16	.01
Change in social closeness	.09 ^f^	1.22	.65 ^fg^	1.22	−.08 ^g^	1.31	6.69 **	.04

b–g similar superscript letters denote a significant difference found in post-hoc pairwise comparisons using Bonferroni correction. * Correlation is significant at the .05 level (2-tailed). ** Correlation is significant at the .01 level (2-tailed).

**Table 3 behavsci-15-00460-t003:** Regression analysis coefficients predicting change in social closeness.

	B	Std. Error	Beta	t
Religiosity	.38	.11	.19	3.35 **
Condition	.52	.17	.17	3.04 **
Change in Stereotypes	−.19	.09	−.12	−2.11 *
Change in Threat	−.21	.07	−.17	−2.81 **

* Correlation is significant at the .05 level (2-tailed). ** Correlation is significant at the .001 level (2-tailed).

**Table 4 behavsci-15-00460-t004:** Means, standard deviations and bivariate correlations for all research variables.

	M	D	1	2	3	4	5
1. Religion	2.02	.49					
2. Common identity	3.37	1.19	−.18 *				
3. Stereotypes	2.97	.74	.09	−.08			
4. Threat	2.49	.86	.21 *	−.07	.35 **		
5. Social distance	2.20	.70	.05	−.17 *	.10	.42 **	
6. Social closeness	2.90	.76	−.07	.30 **	−.39 **	−.13	−.09

* Correlation is significant at the .05 level (2-tailed). ** Correlation is significant at the .01 level (2-tailed).

**Table 5 behavsci-15-00460-t005:** Means, standard deviations and bivariate correlations for change in research variables.

Change in	M	SD	1	2	3	4	5
1. Religiosity ^!^	2.03	.50					
2. Common identity	.35	1.15	.02				
3. Stereotypes	−.19	.80	.08	−.27 **			
4. Threat	−.02	.85	−.08	−.15	.22 **		
5. Social distance	.05	.75	.02	−.05	.16	.32 **	
6. Social closeness	.24	.78	−.05	.36 **	−.34 **	−.14	−.01

** Correlation is significant at the .01 level (2-tailed). ^!^ Not a change score.

**Table 6 behavsci-15-00460-t006:** Pre and post intervention means, SDs of research variables, and repeated measures ANOVA effects by condition.

Condition		Control N = 45		F (1,44)	η2	Inter-Religious Similarity N = 45		F (1,44)	η2	“Start-Up Nation” Identity N = 47		F (1,46)	η2
		**M**	**SD**			**M**	**SD**			**M**	**SD**		
Common identity	Pre	3.02	1.23	3.00	.06	**3.66**	**1.07**	4.38 *	.09	**3.44**	**1.20**	5.90 *	.11
	Post	3.36	1.16			**4.03**	**1.10**			**3.78**	**1.06**		
Stereotypes	Pre	2.83	.70	1.32	.02	**3.03**	**.75**	7.32 *	.14	3.05	.76	.57	.01
	Post	2.71	.80			**2.64**	**.73**			2.97	.68		
Threat	Pre	**2.29**	**.83**	7.30 *	.14	2.41	.77	2.45	.53	**2.75**	**.91**	4.85 *	.95
	Post	**2.66**	**.77**			2.21	.61			**2.53**	**.72**		
Social distance	Pre	2.20	.72	2.53	.05	2.20	.69	.23	.01	2.22	.69	.04	.00
	Post	2.39	.63			2.15	.61			2.23	.59		
Social closeness	Pre	2.96	.78	3.41	.07	**2.79**	**.76**	6.86 *	.13	2.94	.76	3.31	.06
	Post	3.20	.70			**3.11**	**.68**			3.11	.58		

Bold script indicates a significant change * *p* < .05.

## Data Availability

The raw data supporting the conclusions of this article will be made available by the authors on request.

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
