# Peer review of "Abrahamic Family or Start-Up Nation?: Competing Messages of Common Identity and Their Effects on Intergroup Prejudice"

_behavsci, 2025, doi:10.3390/bs15040460_

Round 1

Reviewer 1 Report

Comments and Suggestions for Authors

The study seems relevant to me to address the current dilemmas surrounding the peaceful coexistence of Jewish and Muslim populations in Israel. The approach, methodology, state of the art, cases presented and their discussion seem well elaborated to me.
However, I have two objections that are easy to correct in the study.
1. Improve the title. At first glance it is very long and not understood. By the way, after reading the text it is perfectly understood, but at the beginning it is not. I suggest that it be simplified, trying to be clear, striking and understandable for the reader. It is not necessary to put the three aspects that are highlighted in the title.
2. The study is focused on young Israeli Muslims. It is not made clear in the study, or at least not to the desired extent, why, what is the reason for only including them and not young Israeli Jews. What is the ultimate purpose of the study: to improve the state's educational policies in the matter? To improve the perception of Muslims in their own country, Israel, compared to Jews? It would be good to clarify these points, so as not to generate doubts in the reader.
Otherwise, it seems to me to be a well-constructed study.

Author Response

  1. Improve the title. At first glance it is very long and not understood. By the way, after reading the text it is perfectly understood, but at the beginning it is not. I suggest that it be simplified, trying to be clear, striking and understandable for the reader. It is not necessary to put the three aspects that are highlighted in the title. Suggesting: Abrahamic family or Start-up nation?: competing messages of common identity and their effects on intergroup prejudice.
  2.  The study is focused on young Israeli Muslims. It is not made clear in the study, or at least not to the desired extent, why, what is the reason for only including them and not young Israeli Jews. What is the ultimate purpose of the study: to improve the state's educational policies in the matter? To improve the perception of Muslims in their own country, Israel, compared to Jews? It would be good to clarify these points, so as not to generate doubts in the reader.
    Otherwise, it seems to me to be a well-constructed study.Added: The results of the previous study, showing that unlike Jewish students, Muslim students studying about the similarities between Islam and Judaism increased prejudice, seemed quite surprising. Therefore, we focused this current study on Muslim students, trying to replicate the effect.

Reviewer 2 Report

Comments and Suggestions for Authors

The research report is very fine with the research procedure (design, collection of data, and analysis) given in detail. 

Sufficient reference to related relevant literature has been given and integrated into the formulation of the problem and design of Study 1 and Study 2. Probably, in Line 81. …”the contact hypothesis” could be briefly explained (may be parenthetically).

The research question and findings appear to be quite relevant and it is interesting to note that the findings indicate that comparative reference to current political policies only exacerbates prejudice.

The researchers have appropriately looked into the socio-psychological aspects of the data and referred to a number of explanatory theories such as contact-hypothesis, ITT, and Common In-Group theory. 

The researchers also identified issues in study 1 and attempted to test explanatory assumptions in study 2, which is commendable.

It is interesting to note that out of the 324 seventh grade students, 220 were female. I was wondering why this sample (7th grade) was chosen for study 1. Also why the sample for study 2 was different (mostly college/university students). Probably, I missed some explanation somewhere.

Overall, the research work is excellent with only the following suggestions:

Line 36. Hamas (acronym for arakat al-Muqāwamah al-ʾIslāmiyyah, meaning Islamic opposition movement) OR (acronym for Arabic name, meaning Islamic opposition movement). Including a meaning helps readers to note the real nature and objective of Hamas.

The English is fine, a few suggestions for improvement are given below.

Comments on the Quality of English Language

Line 41. Jewish fundamentalist or Jewish fundamentalists (should it be plural?)

Line 48. Perhaps the comma (,) before “college students” is not necessary.

Line 73. Kindly try to indicate more clearly who the pronouns in this sentence refer to by rephrasing the sentence. I think Shackleford, et.al’s paper title is much clearer “Palestinians and Israelis believe the other’s God encourages intergroup benevolence” in this respect. 

Lines 119-121. Kindly check the sentence structure. Should it be “Since the results of the study on Muslim students…seemed quite surprising”?

Line 122. “a unit”

Line 403.  A comma (,) before religion. “lead us to expect, religion…”

Author Response

It is interesting to note that out of the 324 seventh grade students, 220 were female. I was wondering why this sample (7th grade) was chosen for study 1.

Added: The previous study focused on Seventh grade students (the grade in which Israeli History curriculum instructs teachers to discuss the Relations of Judaism and Islam). Hence, participants were 324 seventh grade students

. Also why the sample for study 2 was different (mostly college/university students). Probably, I missed some explanation somewhere.

Added: We chose a sample of adults since we assumed that the notion an academic pro-fession based common-identity would strike a chord more strongly among participants of working age.

Line 36. Hamas (acronym for Ḥarakat al-Muqāwamah al-ʾIslāmiyyah, meaning Islamic opposition movement) OR (acronym for Arabic name, meaning Islamic opposition movement). Including a meaning helps readers to note the real nature and objective of Hamas.

Changed to (Arabic acronym, meaning Islamic opposition movement)

Line 41. Jewish fundamentalist or Jewish fundamentalists (should it be plural?) done

Line 54. Perhaps the comma (,) before “college students” is not necessary. done

Line 73. Kindly try to indicate more clearly who the pronouns in this sentence refer to by rephrasing the sentence. I think Shackleford, et.al’s paper title is much clearer “Palestinians and Israelis believe the other’s God encourages intergroup benevolence” in this respect. 

Changed to : Muslim Palestinians and Israeli Jews surprisingly tend to assume that outgroup members’ thinking about God encourages intergroup benevolence

Lines 119-121. Kindly check the sentence structure. Should it be “Since the results of the study on Muslim students…seemed quite surprising”?

Changed to: The results of the previous study, showing that unlike Jewish students, Muslim students studying about the similarities between Islam and Judaism increased prejudice, seemed quite surprising. Therefore, we focused this current study on Muslim students, trying to replicate the effect. However, before replicating the study, we sought to account for the negative effects in additional ways.

Line 122. “a unit” changed to: text

Line 403.  A comma (,) before religion. “lead us to expect, religion…”  done.